# The Border Effects of Dry Matter, Photosynthetic Characteristics, and Yield Components of Wheat under Hole Sowing Condition

Yitao Sun [†], Chao Yang [†], Huajun Liang, Yuyan Yang, Kangmin Bu, Yongli Dong and Jiangbo Hai *

College of Agronomy, Northwest A&F University, Yangling, Xianyang 712100, China
* Correspondence: haijiangbo@126.com; Tel.: +86-13389221092
† These authors contributed equally to this work.

**Abstract:** Wheat can be cultivated by hole sowing, but its border effect has not yet been studied. Therefore, we carried out a field experiment from 2021 to 2022 at the Doukou Crop Experimental Demonstration Station (108°52′ E, 34°37′ N) of Northwest A&F University in Jingyang County, Xianyang City, Shaanxi Province, China. The response of dry matter, photosynthetic characteristics, and yield components of wheat to the border effects under the hole sowing method was studied. The results showed specific border effects on each index of five wheat varieties (XN136, XN175, XN527, XN536, and XN765), among which the border effects of XN175 and XN765 were the most significant, with the highest yield. Subsequent correlation analysis revealed that only grain per spike and intercellular carbon dioxide concentration responded negatively to the border effects, and the rest were positively correlated. Finally, we conducted a random forest model analysis of different indicators of wheat varieties with significant border effects. We found that net photosynthetic rate and aboveground dry matter per plant had the most significant impact and contribution to the border effects. In contrast, grain per spike had the most negligible impact on the border effects. Our results fill a gap in the study of the border effects of wheat under hole sowing cultivation for future researchers.

**Keywords:** border effect; hole sowing; wheat; *Triticum aestivum* L.

## 1. Introduction

Due to the extreme changes in global climate and the rapid growth of population, achieving food supply security under limited arable land conditions is a significant challenge in the 21st century [1–4]. Future food security, therefore, requires further increases in crop yields. According to statistics, to meet global food demand, food production will need to increase by 70–100% by 2050, with an annual increase of more than 4 million tons [5–8], and wheat production needs to grow at 1.7% per year [9–12]. Wheat is one of the most important foods for human beings, which is essential and beneficial to human health. How to maximize the benefits of wheat is of great significance to food production and agricultural income [13–15].

Individuals in the border row usually enjoy better conditions to obtain a higher yield, defined as the border (marginal) effect [16,17]. This effect is usually caused by uncultivated space, which is left between adjacent plots for crop management and differentiation of different varieties [18]. Due to more solar energy, better ventilation, and less nutrient competition, the crop growth and yield of the border rows are better than those of the middle rows [19,20]. Therefore, maximizing the border advantage is essential for improving productivity [21].

The cultivation techniques of wheat can regulate wheat tillering, form a reasonable population, enhance the utilization rate of light energy, and have a great impact on coordinating the relationship between source, sink and flow, increasing yield, and improving

quality [22–24]. The sowing method is an important element of cultivation techniques to regulate the growth and development of wheat. Different sowing methods will lead to changes in the structure of wheat population, and therefore the physiological and metabolic processes of plants will change accordingly, affecting the overall growth and development of wheat, and in turn, affecting the yield and quality [25]. The hole sowing cultivation technology of wheat is a high-efficiency agricultural technology integrating rainfall, drought resistance, and efficient utilization of light and heat resources. As a new cultivation technique, it has many excellent characteristics and a good development prospect.

Based on previous research on the effects of different sowing methods and seeding rates on wheat yield and quality, this study further explored the response of different wheat varieties to the border effect of hole sowing. The main purposes are: (1) To explore the response of wheat border effects under the cultivation mode of hole sowing. (2) To explore which wheat varieties are more suitable for hole sowing cultivation. (3) To explore which indicators have significant border effects and the size of the contribution of each indicator to the border effects.

## 2. Materials and Methods

### 2.1. Test Designs and Determination Methods

This experiment was carried out at Doukou Crop Experimental Demonstration Station of Northwest A&F University from October 2021 to June 2022. The experimental demonstration station is located in Xinglong Village, Yunyang Town, Jingyang County, Xianyang City, Shaanxi Province, China, 108°52′ E, 34°37′ N. The average temperature and precipitation in 2021–2022 were 10.89 °C and 17.33 mm, respectively (Figure 1). The soil in the experimental field was loam. The soil organic matter content in the 0–20 cm soil layer of the experimental field was 18.03 g·kg$^{-1}$, the total nitrogen content was 1.31 g·kg$^{-1}$, the available nitrogen content was 86.3 mg·kg$^{-1}$, and the available potassium was 227.48 mg·kg$^{-1}$.

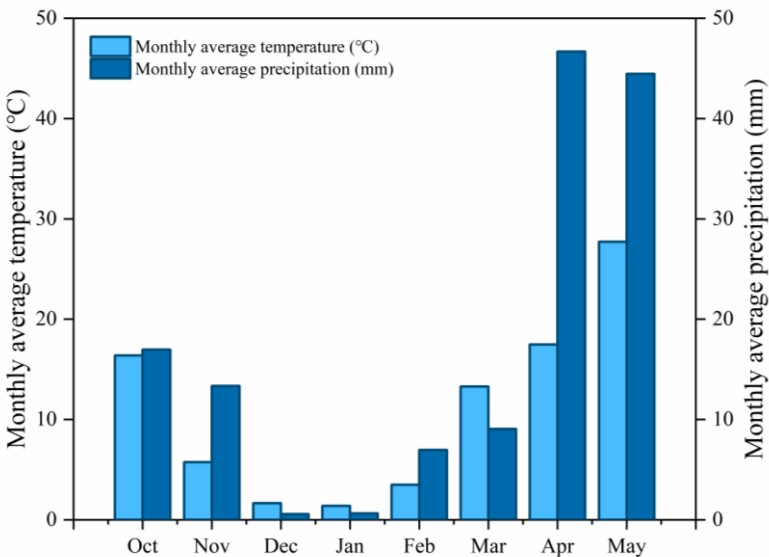

**Figure 1.** Total precipitation and monthly mean temperature during wheat growth stage from October 2021 to June 2022.

Different wheat varieties, 'XN136', 'XN175', 'XN527', 'XN536', and 'XN765', were selected as experimental materials. These five wheat varieties were provided by the College of Agriculture, Northwest A&F University. The main common characteristics were as follows: they were all semi-winter and semi-dwarf varieties, suitable for planting in the Guanzhong irrigation area of Shaanxi Province, had medium tillering ability, high earning rate, fast filling speed, and medium grain plumpness. The main difference was the plant heights. The average plant heights of each variety were: 77.3 cm for 'XN136', 84.1 cm for

'XN175', 77.1 cm for 'XN527', 76.1 cm for 'XN536', and 79.1 cm for 'XN765'. The sowing density was 168.5 kg·ha$^{-1}$, and the sowing amount per hole was 12. In order to ensure the accuracy of the experiment, the sowing method of wheat hole sowing used in this experiment was artificial sowing. Firstly, the furrow opener was used to furrow each plot, where 12 furrows (12 rows) were opened in each plot, and the benchmark was used to mark the points of each row. There were 35 mark points in each row, and 12 grains were sown manually at each mark point. In this experiment, different wheat varieties were used as different treatments, with a total of 5 treatments, three replicates, each plot area of 15 m$^2$, each plot of 12 rows, each row of 35 holes, hole spacing (S1) of 14 cm, and row spacing (S2) of 25 cm (Figure 2). The compound fertilizer (N-P$_2$O$_5$-K$_2$O: 24-15-5) was uniformly applied in the form of base fertilizer at 375 kg·ha$^{-1}$ before tillage. This experiment was sown on 24 October 2021, and harvested on 5 June 2022. Other measures in the experimental field were the same as the requirements of high-yield field cultivation techniques.

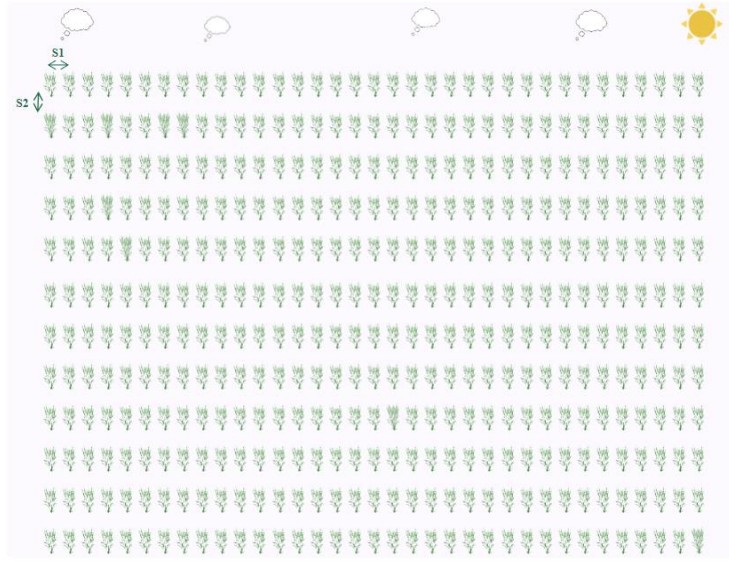

(A)

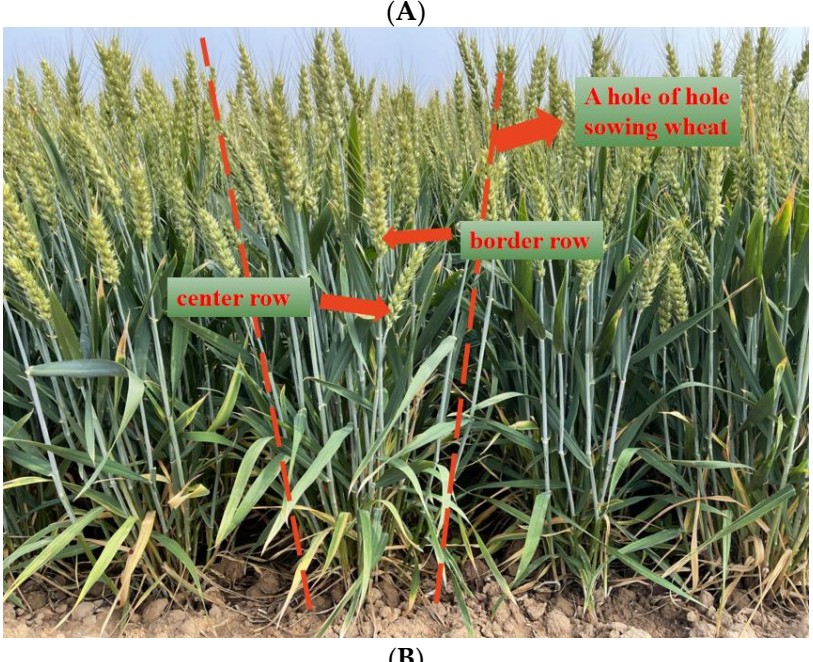

(B)

**Figure 2.** *Cont.*

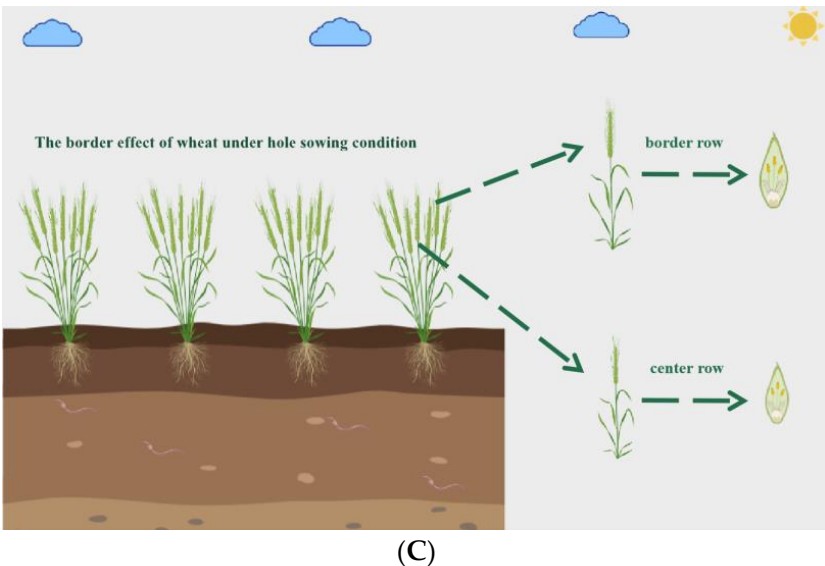

(**C**)

**Figure 2.** Hole sowing planting map of one plot (**A**), hole sowing wheat growth map (**B**), and the border effect map of wheat (**C**).

For our sampling method, to avoid the influence of side rows on the border effects, four holes were randomly sampled in the middle four rows of each plot. Each hole's outer and inner sides were sampled separately, and the average value was used to compare the border effects.

The aboveground dry matter per plant outside and inside each hole of the wheat plant was determined. In the booting stage, heading stage, flowering stage, filling stage, and maturing stage of wheat, four 20 cm plant samples with uniform growth were selected on the outer and inner sides of each hole. After being brought back to the room on the same day, each part was de-enzymed in a 105 °C oven for 30 min, then the temperature was reduced to 60–80 °C, and continued to be dried for about eight hours to make it dry quickly, and then removed. The samples then continued to dry for four hours, weighed again, until the weight was constant at which the final weight was measured.

Wheat plants outside and inside each hole were measured for physiological indexes of photosynthetic characteristics (net photosynthetic rate, stomatal conductance, and intercellular carbon dioxide concentration). In the booting stage, heading stage, flowering stage, and filling stage of wheat, clear and cloudless weather was selected, and the Li-6400 portable photosynthesis system was used to measure the photosynthetic characteristics. Four holes were randomly selected from each plot, and four uniform plant samples were selected on the outside and inside of each hole. The net photosynthetic rate of the middle part of the flag leaf of wheat was measured at 9:00–11:00 a.m. stomatal conductance, and intercellular carbon dioxide concentration. The use of the LI-6400 portable photosynthesis measurement system roughly includes six steps: instrument connection, program loading, instrument correction, data measurement, data transmission, and instrument closing. Before the measurement, we must first see whether the instrument is connected, and then enter the following steps after the instrument is connected. After the power switch is turned on, the instrument begins to install the OPEN program, which takes about ten minutes. The configuration file here must be correctly selected and should be consistent with the type of leaf chamber installed on the head of the IRGA analyzer. Because of the change in the surrounding environmental conditions, the zero point of the instrument changes, and therefore, it must be corrected before use as the data will not be reliable otherwise. When calibrating the instrument, f3 needs to be selected under the OPEN main program interface to enter 'Calib Menu'. After entering the calibration menu, seven secondary menus are displayed on the display screen, among which the first item 'FLOW Meter Zero' (zero adjustment of flowmeter) and the second item 'IRGA Zero' (zero adjustment of infrared

gas analyzer, namely correction of $CO_2$ and $H_2O$ zero points) are necessary operations after each boot. Data measurement is a key step in the use of LI-6400. The data is measured under f4 (New Msmnts, new measurement menu) of the OPEN main interface. Before the experimental data measurement, the $H_2O$ and $CO_2$ control knobs should be adjusted to BYPASS (if the $CO_2$ injection system is used, the $CO_2$ control knob should be adjusted to SCRUB). The measured data is then transmitted to the computer in time.

For yield composition statistics, the number of grains per spike inside each hole was counted. After harvest, the grains were sun-dried to remove impurities. The number of plates used to take each hole outside and inside of a total of 1000 grains were weighed, and repeated three times to calculate the average value of thousand grains. After the wheat matured, the number of effective ears in each plot′s 1 m double-row sample section was counted. Each plot was sampled for 1 m$^2$ of wheat, then threshed with a thresher, dried, weighed with an electronic balance, and calculated for grain yield (kg·ha$^{-1}$).

### 2.2. Statistical Analysis of Data

The border effect (BE%) was calculated as follows according to Wang et al. [15].

$$\text{BE} = \frac{\text{Parameter of border row } - \text{ Parameter of center row}}{\text{Parameter of center row}} \times 100 \qquad (1)$$

Correlation analysis refers to the analysis of two or more correlated variable elements to measure the degree of correlation between the two variable factors.

Random forest regression is a machine learning technique that can create a set of multiple decision trees, aggregate on the set, and rank the predictors according to the correlation between the predictors and the predictions. It is well known that random forest regression techniques can produce highly accurate predictions and handle many input variables without overfitting.

In this study, the outer side of each hole of wheat was used as the border line, and the middle was used as the center line. Correlation analysis and random forest regression analysis were performed according to each index.

Microsoft Office Excel 2021 and SPSS 26.0 were used for data statistical analysis, and RStudio was used for significant difference analysis and picture drawing. The significance level ($p < 0.05$) was used to determine the average difference using the least significant difference test.

### 3. Results

#### 3.1. Border Effects of Yield Components

XN136, XN175, and XN765 have significant border effects (Table 1). The border effects of thousand-grain weight and grain per spike of XN175 were the highest, being 15.1% and 14.2%, respectively, followed by XN765 (11.5%, 12%) and XN136 (6.8%, 5.9%). The effective spikes of XN175 were the largest, at $643 \times 10^4 \cdot \text{ha}^{-1}$, and there was a significant difference between XN175 and XN136, XN536, and XN765. The number of effective spikes per hole of XN175 was the largest, at 23, and the number of effective spikes per plant was 2. The highest yield of XN175 and XN765 was 8587.1 kg·ha$^{-1}$ and 8558.6 kg·ha$^{-1}$, respectively.

#### 3.2. Border Effects of Dry Matter

The aboveground dry matter of wheat at different stages (booting stage, heading stage, flowering stage, filling stage, and maturing stage) was measured. The border effect was analyzed (Figure 3). It can be seen from the figure that the maximum dry matter mass of the five varieties in different stages was at the outer row of wheat, which is the maturing stage of XN175, with a value of 13.17 g/plant. The minimum value of dry matter was found in wheat inline, also wheat XN175, which appeared at booting stage and was 3.37 g/plant. It can be seen that the dry matter of the aboveground plants of the five varieties showed a particular border effect, among which XN136 only had significant differences in the dry matter border effect of the aboveground plants at the heading stage

and flowering stage. The dry matter border effect of XN175 in the booting stage and filling stage was significantly different, and the dry matter border effect in the heading stage and the maturing stage was significantly different. The dry matter border effect of XN527 in the five stages was insignificant. XN536 only significantly differed in the dry matter border effect of aboveground dry matter per plant in the flowering stage. The border effect of dry matter per plant above ground of XN765 was significantly different in each stage. It can be seen that the aboveground dry matter of the two wheat varieties, XN175 and XN765, had a significant border effect under hole sowing conditions.

**Table 1.** Border effects of yield and yield components of wheat at maturity stage in 2021–2022 ($p \leq 0.05$, significant difference when the outline and inline characters of the same variety are completely different; $p > 0.05$, no significant difference when the same or more letters are used).

| Variety | Location | Thousand-Grain Weight (g) | Grain Per Spike | Effective Spikes Per Hole | Effective Spikes ($\times 10^4 \cdot ha^{-1}$) | Yield ($kg \cdot ha^{-1}$) | Thousand-Grain Weight (BE%) | Grain Per Spike (BE%) |
|---|---|---|---|---|---|---|---|---|
| XN136 | outer | 56.41 ab | 65.67 b | 18 a | 506.7 c | 8358.3 ab | 6.8% | 5.9% |
|  | inner | 52.83 d | 62 b |  |  |  |  |  |
| XN175 | outer | 57.66 a | 75 a | 23 a | 643 a | 8587.1 a | 15.1% | 14.2% |
|  | inner | 50.1 e | 65.67 b |  |  |  |  |  |
| XN527 | outer | 53.83 b | 53 c | 22 a | 604 ab | 7474.9 b | −2.8% | 1.3% |
|  | inner | 55.39 abcd | 52.33 c |  |  |  |  |  |
| XN536 | outer | 55.57 abc | 42 e | 19 a | 539 c | 8085.3 ab | 3.1% | 3.3% |
|  | inner | 53.9 b | 40.67 e |  |  |  |  |  |
| XN765 | outer | 53.22 cd | 56 c | 20 a | 570 bc | 8558.6 a | 11.5% | 12% |
|  | inner | 47.71 e | 50 d |  |  |  |  |  |

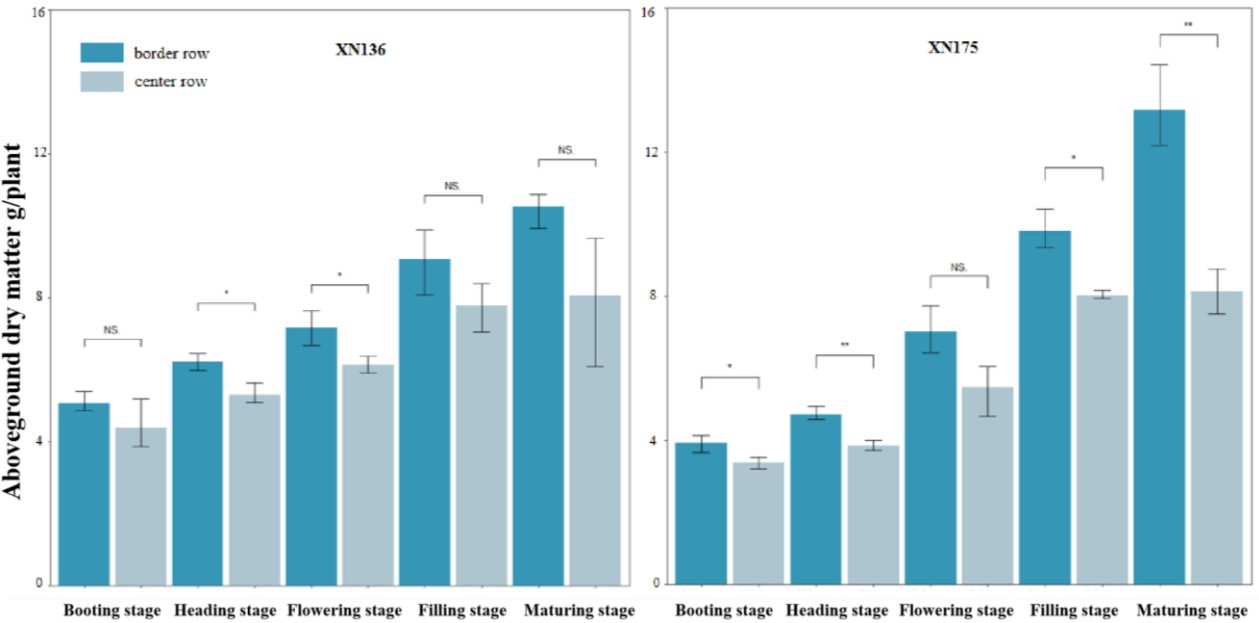

**Figure 3.** *Cont.*

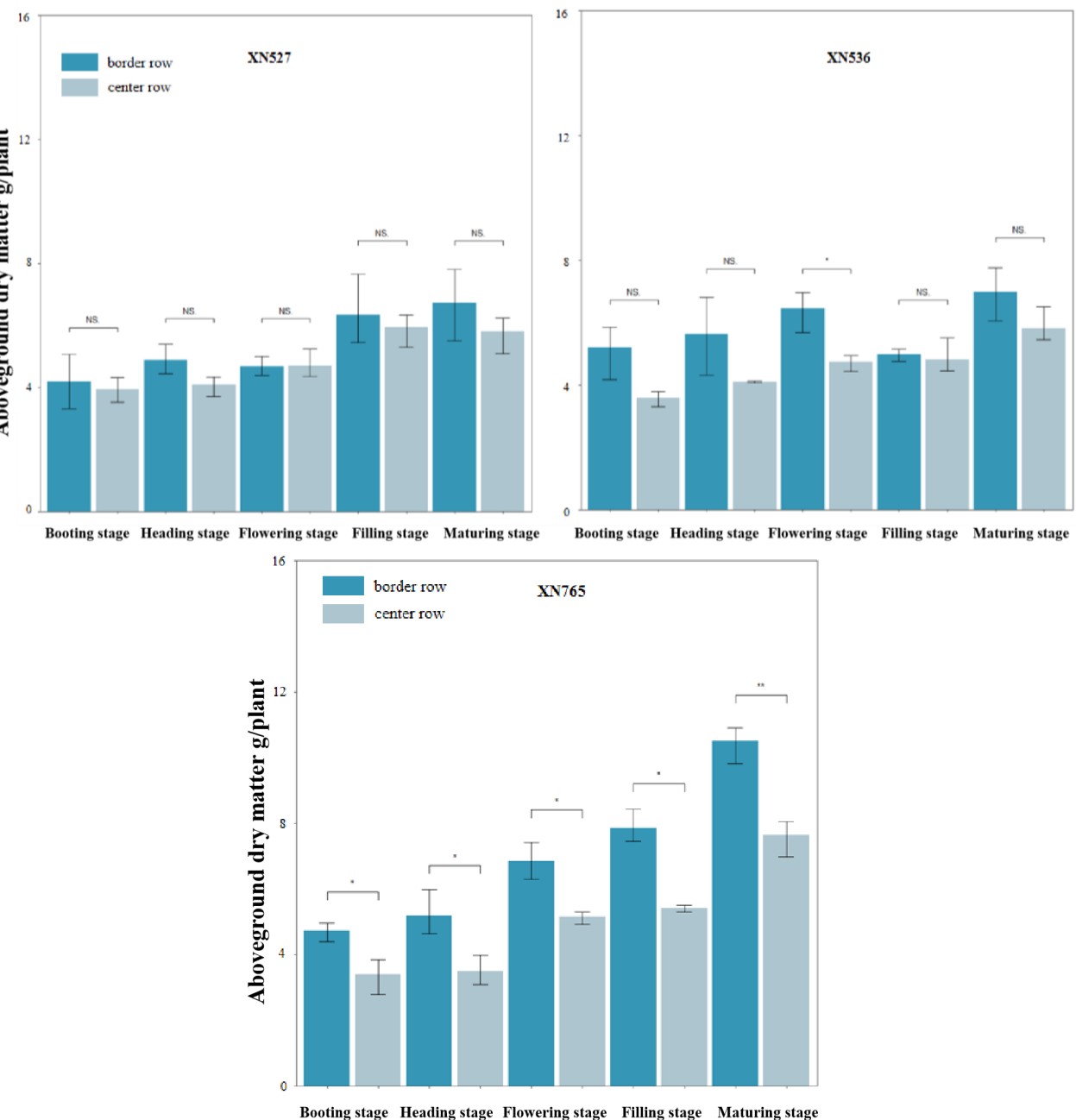

**Figure 3.** Border effect of aboveground dry matter of wheat in different stages. (*: $p \leq 0.05$, **: $p \leq 0.01$, and NS: non-significant (ANOVA)).

### 3.3. Border Effects of Photosynthetic Characteristics

The photosynthetic characteristics of wheat at different stages (booting stage, heading stage, flowering stage, and filling stage) were measured and the border effects were analyzed.

#### 3.3.1. Border Effects of Net Photosynthetic Rate

The border effect of the net photosynthetic rate in different stages of wheat was analyzed (Figure 4). It can be seen that the net photosynthetic rate of the five varieties in different stages had a specific border advantage. The net photosynthetic rate of the five wheat varieties reached the peak at the heading stage, and reached the lowest value at the filling stage. Among them, only the border effects of XN175 and XN765 were significantly different in particular stages, indicating that these two varieties could exert obvious border

effect advantages under hole sowing conditions. Although the remaining three varieties have a certain border effect, the difference was not significant.

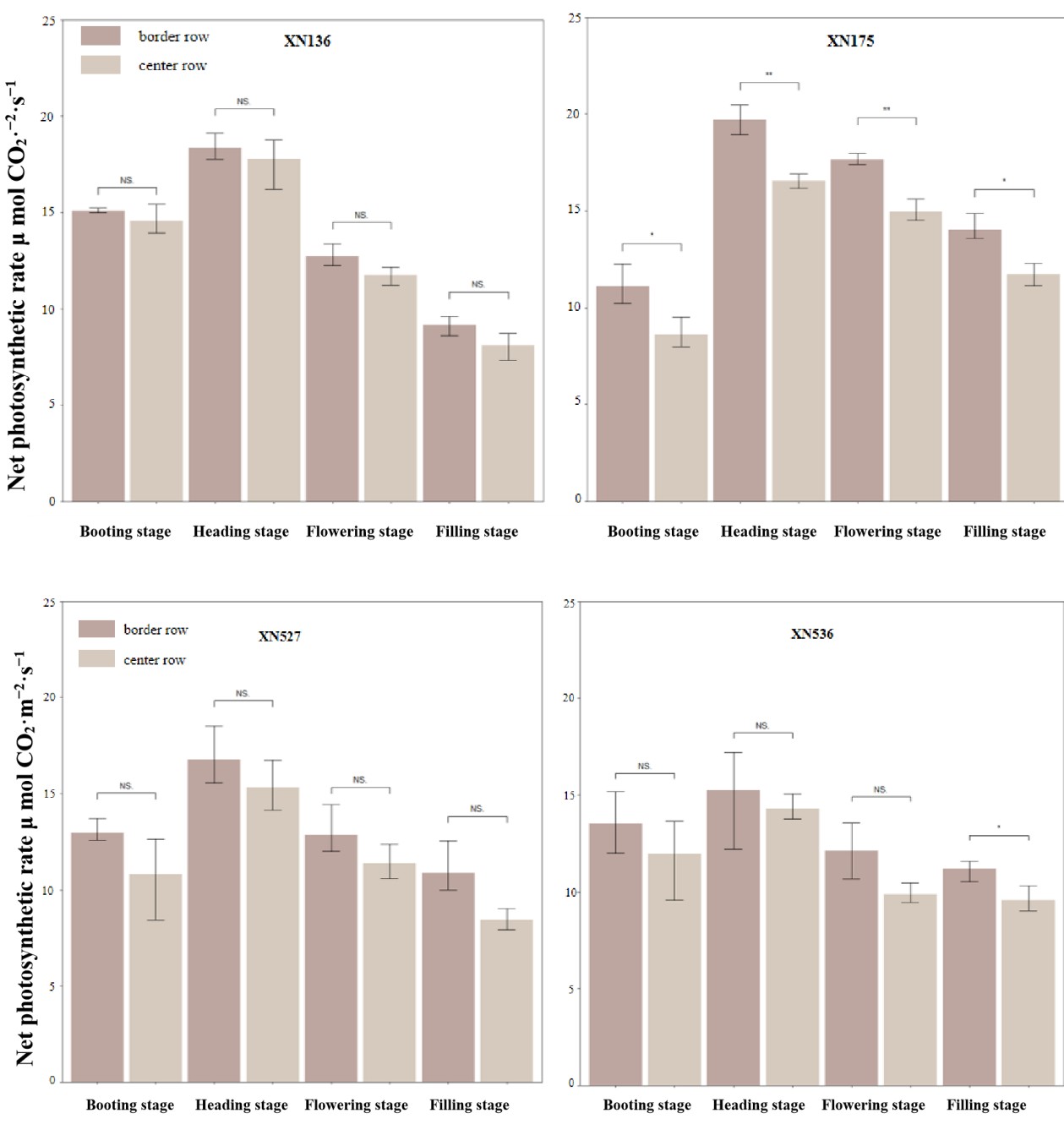

**Figure 4.** *Cont.*

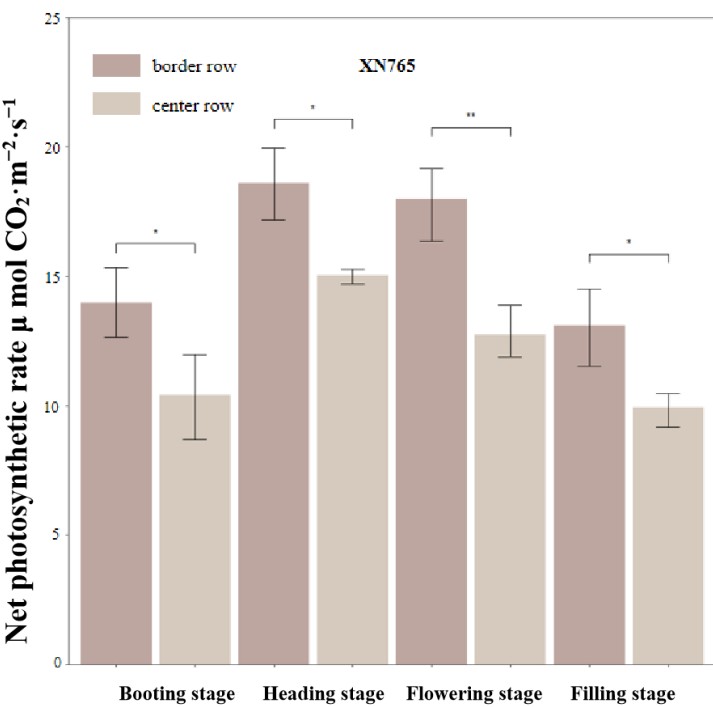

**Figure 4.** Border effect of net photosynthetic rate of wheat in different stages. (*: $p \leq 0.05$, **: $p \leq 0.01$, and NS: non-significant (ANOVA)).

3.3.2. Border Effects of Stomatal Conductance

The border effect of stomatal conductance in different stages of wheat was analyzed (Figure 5). The stomatal conductance of the three wheat varieties, XN136, XN527, and XN536, had a certain border effect in each stage, but the difference was not significant and all three varieties reached the maximum at the booting stage. This showed an overall downward trend. On the contrary, XN175 and XN765 were significantly different in different stages, where both showed an upward trend from the booting stage to the filling stage, and reached the maximum at the filling stage. It can be seen that XN175 and XN765 can play a greater advantage than the other three varieties under hole sowing conditions.

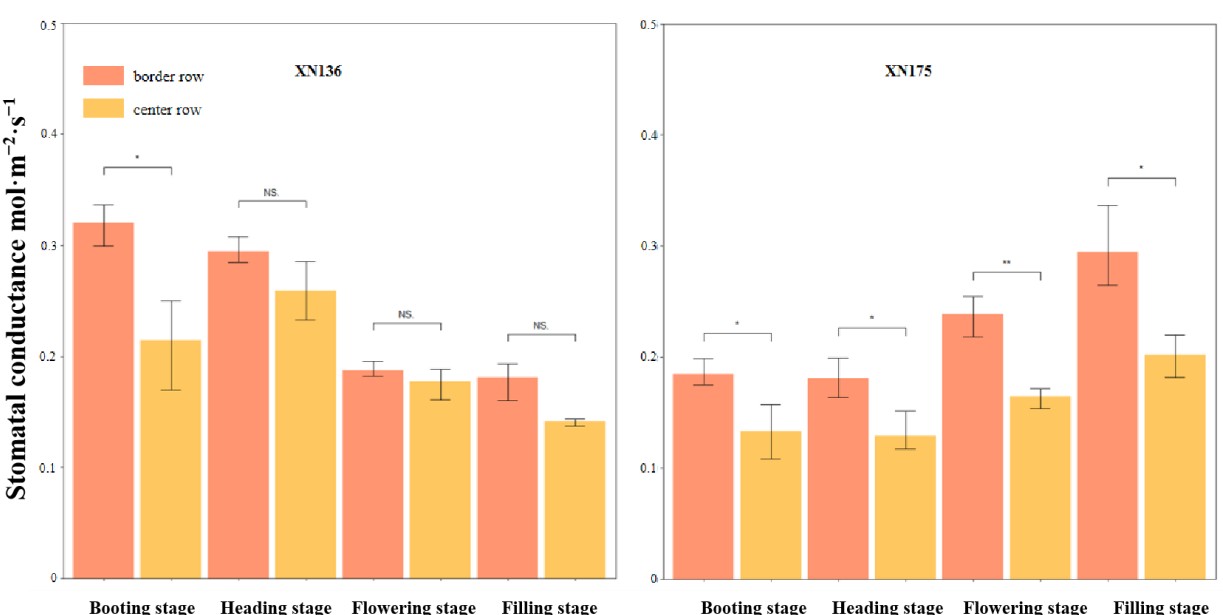

**Figure 5.** *Cont.*

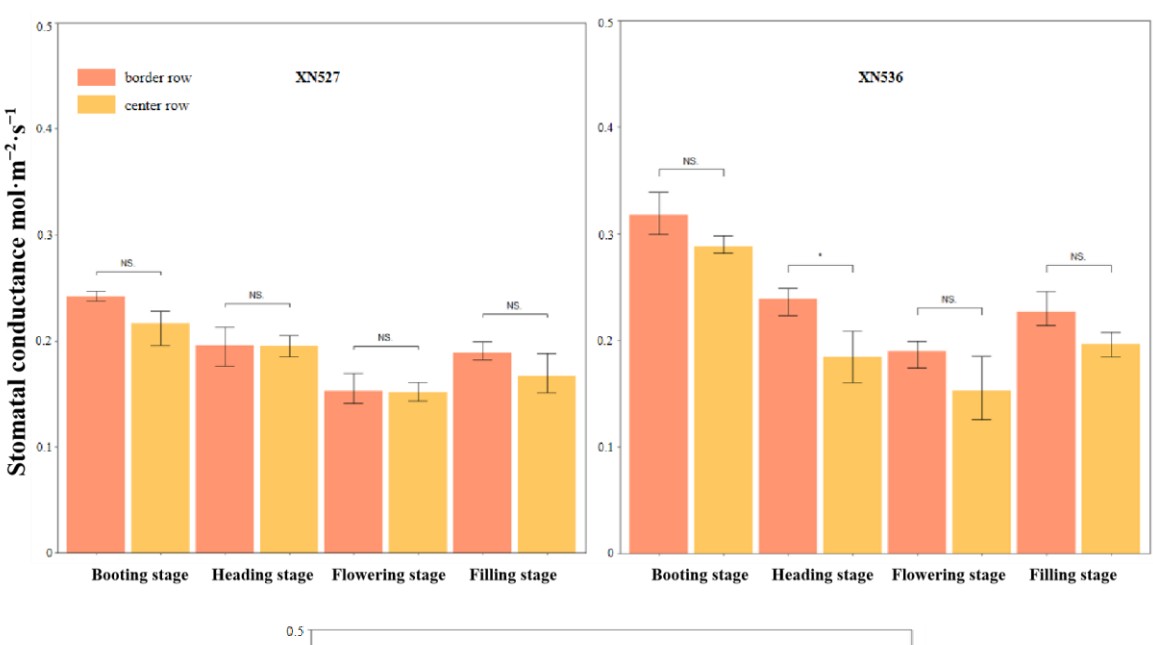

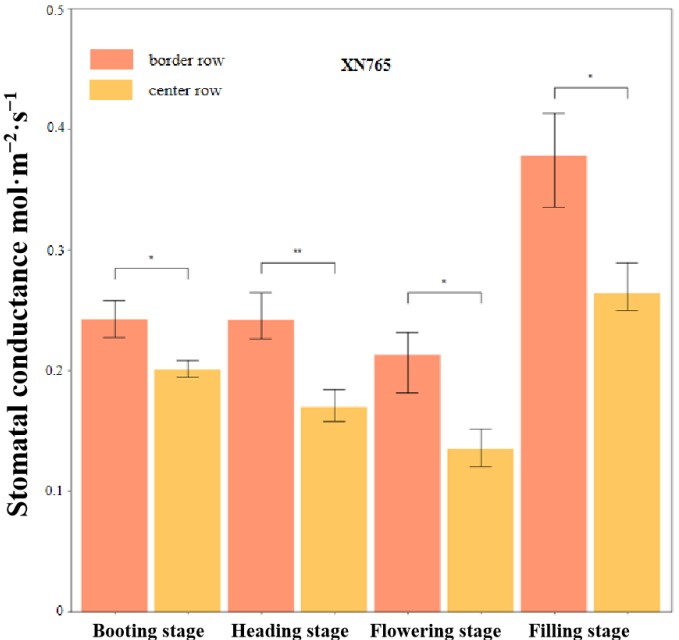

**Figure 5.** Border effects of stomatal conductance of wheat in different stages. (*: $p \leq 0.05$, **: $p \leq 0.01$, and NS: non-significant (ANOVA)).

### 3.3.3. Border Effects of Intercellular Carbon Dioxide Concentration

The border effects of wheat intercellular carbon dioxide concentration at different stages were analyzed (Figure 6). The intercellular carbon dioxide concentration of XN136, XN527, and XN536 was relatively stable in different stages. In contrast, the intercellular carbon dioxide concentration of XN175 and XN765 fluctuated wildly and peaked at the filling stage. Overall, the intercellular carbon dioxide concentration of XN175 and XN765 under hole sowing conditions had a more significant border effect.

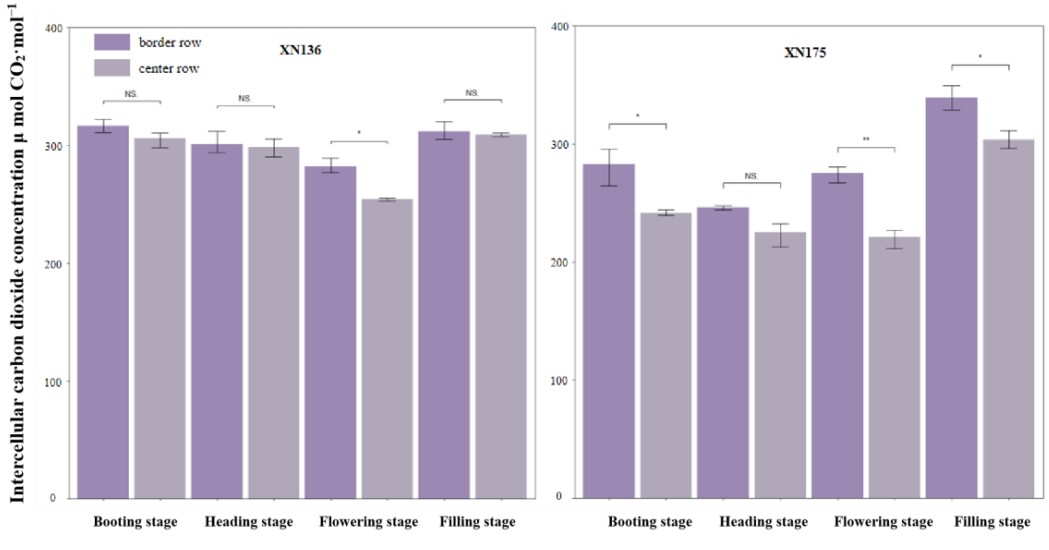

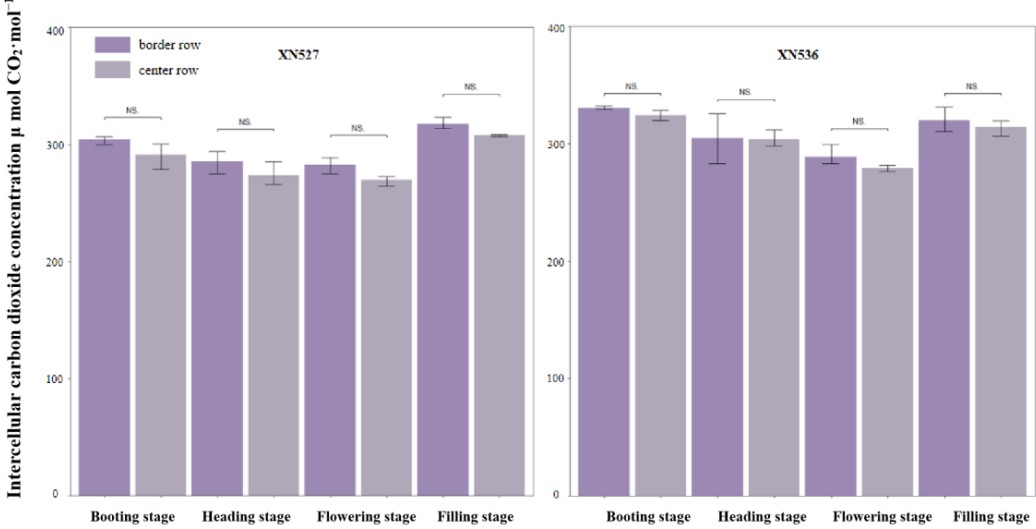

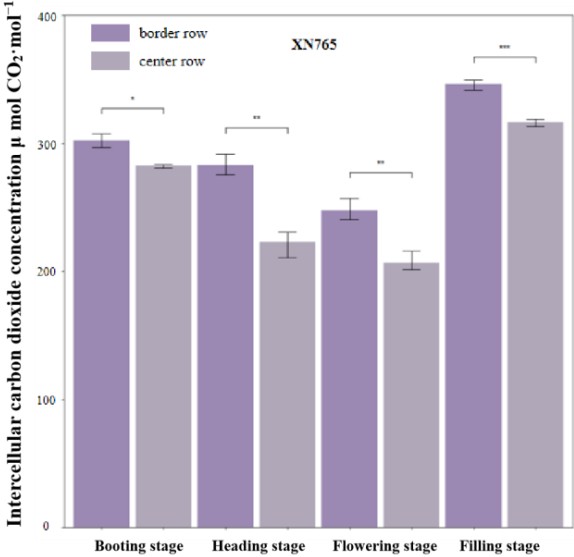

**Figure 6.** Border effects of intercellular carbon dioxide concentration of wheat in different stages. (*: $p \leq 0.05$, **: $p \leq 0.01$, ***: $p \leq 0.001$, and NS: non-significant (ANOVA)).

### 3.4. Correlation Analysis of Different Indexes of Wheat

Correlation analysis was performed on net photosynthetic rate, stomatal conductance, intercellular carbon dioxide concentration, dry matter per plant, thousand-grain weight, and grain per spike of the five wheat varieties (Figure 7). In the figure, different colors represent positive and negative correlations, and the color depth represents the correlation size. The bluer the color, the greater the positive correlation coefficient; the redder the color, the greater the negative correlation coefficient. It was found that only grain per spike and intercellular carbon dioxide concentration responded negatively to the border effect, and the rest were positively correlated. Among them, grain per spike and aboveground dry matter per plant, stomatal conductance and intercellular carbon dioxide concentration, thousand grain weight, and intercellular carbon dioxide concentration had significant positive correlations with the border effect. There was a significant positive correlation between net photosynthetic rate and aboveground dry matter per plant, which was the most important factor affecting the maximum border effect of wheat under hole sowing conditions.

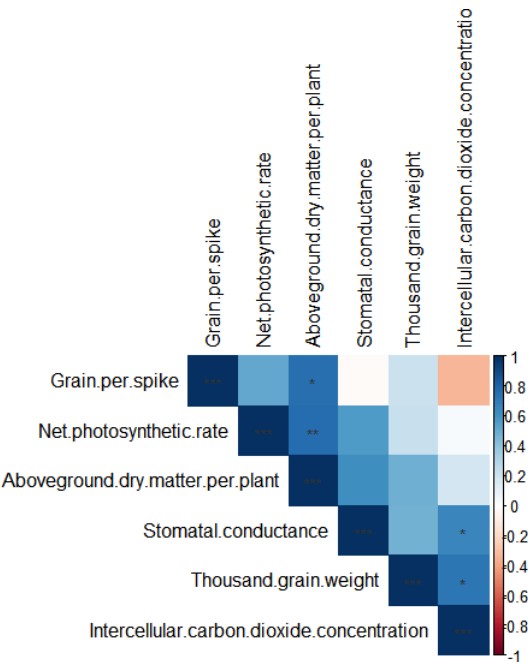

**Figure 7.** Correlation analysis of different wheat indexes (X axis and Y axis represent different indexes, r values in the Figure in different colors, *: $p \leq 0.05$, **: $p \leq 0.01$).

### 3.5. The Contribution of Different Indicators to Its Significant Border Effects

'Mean decrease Gini' is used to calculate the influence of each variable on the heterogeneity of observations at each node of the classification tree, and to compare the importance of variables. The larger the value, the greater the variable's importance is.

Only XN175 and XN765 showed significant differences in the border effects of different indicators. Therefore, random forest model analysis was performed on the state, net photosynthetic rate, stomatal conductance, intercellular carbon dioxide concentration, aboveground dry matter per plant, thousand-grain weight, and grain per spike of these two varieties (Figure 8). As can be seen from Figure 8, for 'mean decrease Gini', aboveground dry matter per plant had the most significant response to the border effect, and grain per spike was the smallest.

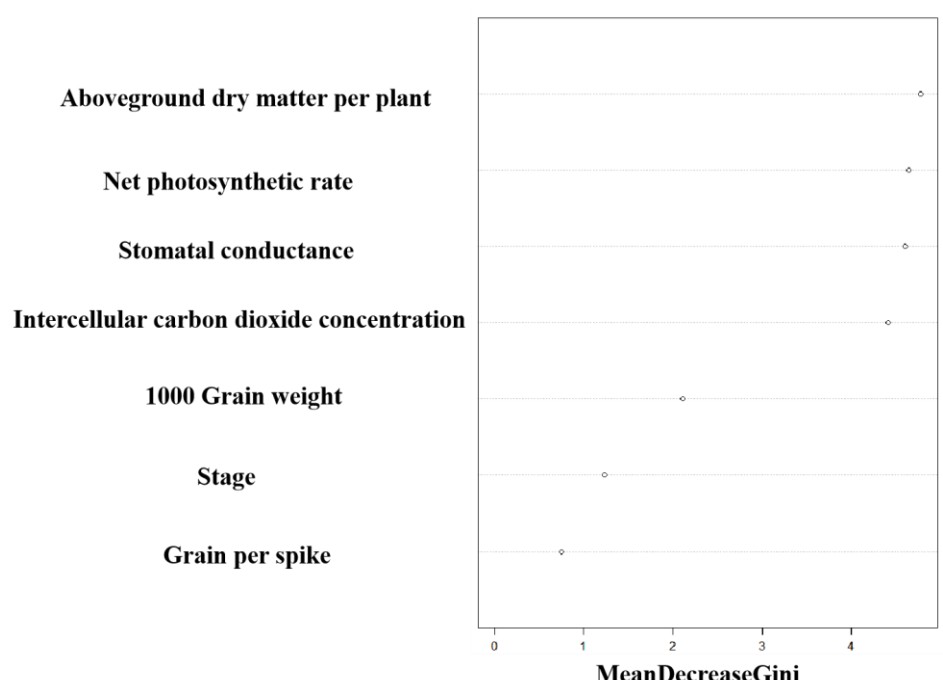

**Figure 8.** Response of different indexes of wheat cultivars, XN175 and XN765, to border effect (Multi-class area under the curve: 0.9722).

### 3.6. Difference Analysis of Each Index between Different Wheat Varieties

The photosynthetic characteristics of wheat during the filling stage is the key stage to determine the yield, and about 80% of the nutrients are transported to the wheat grain for accumulation in the middle filling stage. In order to select the most suitable wheat variety for hole sowing among the five varieties, the indexes of wheat filling stage and final yield of the five varieties were compared (Table 2). Among the dry matter per plant, XN175 had the highest value of 8.92 g. Among the photosynthetic characteristics, XN175 had the highest net photosynthetic rate of 12.87 $\mu$ mol $CO_2 \cdot m^{-2} \cdot s^{-1}$. The stomatal conductance of XN765 was the largest, which was 0.37 mol$\cdot m^{-2} \cdot s^{-1}$. In the final yield, XN175 was also the highest, which was 8587.1 kg$\cdot ha^{-1}$. Combined with previous results, XN175 and XN765 has been demonstrated to play a greater advantage under hole sowing conditions.

**Table 2.** Differences in dry matter, photosynthetic characteristics, and final yield of different wheat varieties during grain filling stage from 2021 to 2022. ($p \leq 0.05$, significant difference when the outline and inline characters of the same variety are completely different; $p > 0.05$, no significant difference when the same or more letters are used).

| Variety | Dry Matter Per Plant (g) | Net Photosynthetic Rate ($\mu$ mol $CO_2 \cdot m^{-2} \cdot s^{-1}$) | Stomatal Conductance (mol$\cdot m^{-2} \cdot s^{-1}$) | Intercellular Carbon Dioxide Concentration ($\mu$ mol $CO_2 \cdot mol^{-1}$) | Yield (kg$\cdot ha^{-1}$) |
|---------|--------------------------|--------------------------------------------------------|-----------------------------------------|----------------------------------------------------|---------------------------|
| XN136 | 8.42 a | 8.64 c | 0.17 c | 337.18 a | 8358.3 ab |
| XN175 | 8.92 a | 12.87 a | 0.36 ab | 336.4 a | 8587.1 a |
| XN527 | 6.15 bc | 9.65 bc | 0.26 bc | 344.22 a | 7474.9 b |
| XN536 | 4.90 c | 10.4 bc | 0.29 ab | 349.9 a | 8085.3 ab |
| XN765 | 6.63 b | 11.51 ab | 0.37 a | 342.88 a | 8558.6 a |

### 4. Discussion

The study of the benefits of crop borders usually has two purposes: (1) to avoid the overestimation of crop yields in field trials, and (2) to increase crop productivity by using skip row and rectangular planting patterns. Increasing the dry weight of stubble and non-structural carbohydrate accumulation at harvest of main crops may be an essential strategy

for developing high-yield planting practices in rice regeneration systems, by studying the border effects of principal crops and regenerated crops in rice regeneration systems [15]. By measuring the border effect of the rectangular geometry transplanted with wide and narrow hill spacing, by quantifying the size and shape of the hybrid rice planting plot, it was found that for plots with a larger rectangular shape and smaller plot size, the yield estimation will be higher [17]. Maize hybrids have the potential to increase yield through the intercropping system, and through the study of the border effect of maize hybrid intercropping, it was shown that the land equivalent ratio is affected by the use of intercropping hybrids and seasonal climate change [18]. The border effect on the yield of regenerated crops in a mechanized rice regeneration system have also been studied [22]. They proved that the rolling of main crops during mechanical harvesting had a border effect on the yield of the non-rolling zone, thereby reducing the yield loss of regenerated crops.

Our research group has proved that the hole sowing method has an excellent effect on the growth characteristics of wheat, through the comparative test of wheat hole sowing and traditional sowing methods. For example, (1) the effects of different sowing methods on wheat yield and quality was studied by Wu et al. [26], who explored the effects of different sowing methods (drill sowing, wide sowing, and hole sowing) on wheat yield and quality by applying nitrogen fertilizer to wheat 'Xinong 805'. The results showed that the hole sowing treatment increased the flag leaf area of wheat. The application of nitrogen fertilizer increased the dry matter quality of the above-ground part of the wheat in the hole sowing treatment, and the actual yield of the wheat in the hole sowing treatment was the highest, of up to 7430 kg·ha$^{-1}$. The basic seedlings, biomass, and harvest index of wheat under different sowing methods were significantly different. Under the application of nitrogen fertilizer, the storage material transfer amount and contribution rate of each vegetative organ in the hole sowing treatment were the highest. In addition, the hole sowing treatment under topdressing nitrogen fertilizer increased the volume mass, sedimentation value, protein mass fraction, hardness, stability time, tensile area, elongation, and maximum tensile resistance of the grain. For the effects of different sowing methods and sowing rates on wheat yield and quality (2), Qi et al. [27] studied the effects of different sowing methods and sowing rates on grain yield, yield components, protein content, component content, and processing quality of winter wheat. Using high-quality and high-yield winter wheat 'Xinong20' as material, three different sowing methods (drill sowing, wide sowing, and hole sowing) and four different sowing rates (112.5, 150, 187.5, and 225 kg·ha$^{-1}$) were set up for the experiment. Hole sowing is beneficial to the improvement of protein and its components content and processing quality. Increasing the appropriate sowing rate can increase the content of protein and its components.

Previously, no scholars have studied the border effect of wheat under hole-sowing conditions and the main factors affecting its border effect. In this study, the traits of five wheat varieties showed different border effects under the hole-sowing cultivation method. However, only the different indicators of XN175 and XN765 have significant differences. In dry matter, XN175 had significant difference in the boundary effect of dry matter per plant above ground at booting stage and filling stage, while XN765 had significant difference in the boundary effect of dry matter per plant above ground at maturing stage, and the other four stages had significant difference. In the photosynthetic characteristics, the net photosynthetic rate boundary effect of XN175 and XN765 in each stage was significantly different. The stomatal conductance of XN175 and XN765 increased with the growth stage, and had significant boundary effects at different growth stages of wheat. The intercellular carbon dioxide concentration of XN175 was significantly different at booting stage, flowering stage, and filling stage, and the intercellular carbon dioxide concentration of XN765 was significantly different at each stage. It can be seen that these two varieties play an advantage over the other three varieties under the cultivation method of hole sowing, and have higher wheat yields. XN175 and XN765 may be more suitable for bunch planting than the other three varieties, and have significant border effects. This study only studied the border effect of wheat under the condition of hole sowing from the same

sowing density. In the future, it is necessary to further study the boundary effect response of sowing density to wheat. At present, there are many cultivation methods, but the traits of different wheat varieties under various cultivation methods should be different. Therefore, it is necessary to establish a model to match the best cultivation methods for wheat in the future.

Through the correlation analysis of different indexes of wheat, it can be found that only grain per spike and intercellular carbon dioxide concentration were negatively correlated with the border effect of wheat under hole sowing conditions, while the rest were positively correlated. Through further random forest model analysis of XN175 and XN765 wheat varieties with significant border effects of each index, it can be found that net photosynthetic rate and aboveground dry matter per plant have the greatest influence on the significant border effect. In contrast, grain per spike has a minor influence on the significant border effect.

## 5. Conclusions

Under the cultivation mode of hole sowing, different wheat varieties have specific border effects. The varieties with the most significant border effect may be more suitable for hole sowing than other varieties. Under the warm temperate continental monsoon climate conditions, such as those found in the Guanzhong irrigation area in Shaanxi Province, wheat suitable for hole sowing, as a sowing method, can maximize its performance and obtain higher yield. According to our experiment, 'XN175' and 'XN765' had more significant border effects than other varieties under hole sowing conditions. Therefore, 'XN175' and 'XN765' were more suitable for sowing under hole sowing conditions than the other three varieties, and should be fully considered in the popularization and application of hole sowing. Our results fill the gap in the study of the border effect of wheat under the hole-sowing cultivation method. Readers can obtain exciting information from the data analysis of this study, which provides a valuable reference and help for future researchers.

**Author Contributions:** Y.S.: Investigation, Methodology, Writing—original draft; C.Y.: Writing-original draft; H.L.: Methodology; Y.Y.: Writing—review & editing; K.B.: Writing—review & editing; Y.D.: Methodology; and J.H.: Project administration, Funding acquisition. All authors have read and agreed to the published version of the manuscript.

**Funding:** The research was financially supported by the Ecological Security and Bioremediation Mechanism of Saline-alkali Soil Improvement in the Middle Yellow River (No. DL2021172002L).

**Institutional Review Board Statement:** Not applicable.

**Data Availability Statement:** Not applicable.

**Acknowledgments:** The authors express their special gratitude to the funding source for the financial assistance and are also thankful to the anonymous reviewers for their constructive and valuable comments on earlier versions of this research article.

**Conflicts of Interest:** The authors declare no conflict of interest.

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
