# Peer review of "The Border Effects of Dry Matter, Photosynthetic Characteristics, and Yield Components of Wheat under Hole Sowing Condition"

_agronomy, doi:10.3390/agronomy13030766_

Round 1

Reviewer 1 Report

Well conducted research. Article approved with minor correction suggestions.

Best regards,

Author Response

Response to comments

Dear Editor and Reviewer

We are grateful to the Journal of Agronomy for providing us with the valuable opportunity to revise and modify our manuscript. We would like to thank the reviewers for their careful review and comments, and for your hard work. All the comments have been carefully revised and highlighted in red color for your convenience.

We have tried our best to improve the manuscript and have made comprehensive changes in this paper. After reading the latest version of our manuscript, we hope you will have a better understanding of this manuscript and look forward to your evaluation. We earnestly appreciate the reviewers’ work and we hope that the changes and corrections will meet with your approval. Once again, thank you very much for your comments and suggestions.

Yours sincerely,

The author

Reviewer 2 Report

General comments

Generally the idea of hole sowing is interesting and the results presented in the manuscript shows expected effects of border effects.

In my opinion, the main problem of this manuscript is the lack of comparison of the effects of the new method of sowing with traditional sowing.

It is not clear what benefits the use of wheat sowing in accordance with the proposed method may bring. In the introduction, the authors rightly emphasize the need to increase yields and the mass of food produced. However, we do not know from the work whether the sowing method described will ensure an increase in yield. Is it a better method than the ones used so far? If so, why?

In my opinion this manuscript should not be publish in the journal Agronomy (MDPI).

Detail comments

Introduction

The issue of sowing method and border effects are correctly presented in the introduction but there is few number of reference used.

Materials and methods

Generally, this section is very well presented. I suggest minor corrections and additions:

Some mistakes are pointed in the manuscript attached.

Results

Generally, this section is correctly presented. Some mistakes and inaccuracies are pointed in the manuscript attached.

Discussion

The discussion of the results is very poor. A very small number of literature items were used. In open access, information on the boundary effect and the proximity effect in cereals, including wheat, can be obtained. However, the authors did not use the available literature.

Conclusions.

The conclusions are drafted correctly according to the results obtained.

Another technical remarks are pointed in the manuscript attached.

Author Response

Response to comments

Dear Reviewer

  We are grateful to the Journal of Agronomy for providing us with the valuable opportunity to revise and modify our manuscript. We would like to thank the reviewers for their careful review and comments, and for your hard work. All the comments have been carefully revised and highlighted in red color for your convenience. We shall look forward to hearing from you and hope that you will consider our work again and give us the opportunity to communicate further with the reviewers as needed.

Response to reviewer comments:

  1. In my opinion, the main problem of this manuscript is the lack of comparison of the effects of the new method of sowing with traditional sowing. It is not clear what benefits the use of wheat sowing in accordance with the proposed method may bring. In the introduction, the authors rightly emphasize the need to increase yields and the mass of food produced. However, we do not know from the work whether the sowing method described will ensure an increase in yield. Is it a better method than the ones used so far? If so, why?

Response: Thanks for pointing out the flaw in the manuscript, Our previous work and research have compared the hole sowing of wheat with the traditional sowing method, and also explored the effects of different sowing methods and sowing rates on the quality and yield of wheat. The results show that the hole sowing has a better effect on some indicators of wheat growth characteristics and can effectively increase the yield. We have added previous experiments and results to the discussion section of the article and cited references to published articles.

Because the previous experiment was only the same wheat variety, this experiment was based on the previous experiment to further explore the differences of different varieties of wheat under the condition of hole sowing and whether it had significantborder effect, so as to provide valuable reference and help for future researchers in this field. “For example : (1) the effects of different sowing methods on wheat yield and quality : Wu et al. [29] explored the effects of different sowing methods ( drill sowing, wide sowing and hole sowing ) on wheat yield and quality by applying nitrogen fertilizer to wheat ' Xinong 805 '. The results showed that the hole sowing treatment increased the flag leaf area of wheat. The application of nitrogen fertilizer increased the dry matter quality of the above-ground part of the wheat in the hole sowing treatment, and the actual yield of the wheat in the hole sowing treatment was the highest, up to 7430kg·hm-2. The basic seedlings, biomass and harvest index of wheat under different sowing methods were significantly different. Under the application of nitrogen fertilizer, the storage material transfer amount and contribution rate of each vegetative organ in the hole sowing treatment were the highest ; in addition, the hole sowing treatment under topdressing nitrogen fertilizer increased the volume mass, sedimentation value, protein mass fraction, hardness, stability time, tensile area, elongation and maximum tensile resistance of the grain. Effects of different sowing methods and sowing rates on wheat yield and quality : (2) Qi et al. [30] studied the effects of different sowing methods and sowing rates on grain yield, yield components, protein content and its component content and processing quality of winter wheat. Using high-quality and high-yield winter wheat ' Xinong20 ' as material, three different sowing methods ( conventional drilling, wide sowing and hole sowing ) and four different sowing rates ( 112.5, 150, 187.5, and 225 kg·hm-2 ) were set up for the experiment. The results showed that compared with conventional drilling and wide sowing, the protein content of hole sowing was significantly increased by 6.34 % and 2.83 %, and the ratio of albumin, globulin, glutenin and glutenin was the highest. Hole sowing is beneficial to the improvement of protein and its components content and processing quality. Increasing the appropriate sowing rate can increase the content of protein and its components. ”(Lines 640-364) 

  1. Introduction

The issue of sowing method and border effects are correctly presented in the introduction but there is few number of reference used.

Response: Thank you very much for pointing out the problem of our manuscript, we added the references to the text. “Wheat is one of the most important food for human beings and feeds billions of people around the world. In addition to being the main source of rations, wheat is also an important dietary component for most people in developing countries, which is essential and beneficial to human health [10-13].Although global wheat production is not large, accounting for less than 1 % of global wheat production, it is concentrated in relatively small geographical areas and can be regarded as a major cereal crop, making a significant contribution to food production and agricultural income [14-16].”(Lines 33-40) 

“At present, the research of wheat is mainly focused on breeding, and the supporting cultivation techniques are not good. Therefore, it is of great significance to study the cultivation techniques of high quality wheat. Cultivation techniques can regulate wheat tillering, form a reasonable population, enhance the utilization rate of light energy, and have a great impact on coordinating the relationship between source, sink and flow, increasing yield and improving quality [22-24]. Sowing method is an important cultivation technique to regulate the growth and development of wheat. Different sowing methods will lead to changes in the structure of wheat population, so that the physiological and metabolic processes of plants will change accordingly, affecting the overall growth and development of wheat, and then affecting the yield and quality [25-28]. ”(Lines 48-58) 

  1. Materials and methods

Generally, this section is very well presented. I suggest minor corrections and additions: Some mistakes are pointed in the manuscript attached

Response: Thanks for pointing out the flaw in the manuscript, We have revised this part.

  1. Results

Generally, this section is correctly presented. Some mistakes and inaccuracies are pointed in the manuscript attached.

Response: Thanks for pointing out the flaw in the manuscript, We have revised the errors in the text according to your suggestions and reimproved the quality of pictures and tables.

  1. Discussion

The discussion of the results is very poor. A very small number of literature items were used. In open access, information on the boundary effect and the proximity effect in cereals, including wheat, can be obtained. However, the authors did not use the available literature.

Response: Thanks for pointing out the flaw in the manuscript, We have made a lot of modifications to the discussion section and cited our previous research and other scholars ' research. (Lines 321-364).

In conclusion, we have tried our best to improve the manuscript and have made comprehensive changes in this paper. After reading the latest version of our manuscript, we hope you will have a better understanding of this manuscript and look forward to your evaluation. We earnestly appreciate the reviewers’ work and we hope that the changes and corrections will meet with your approval. Once again, thank you very much for your comments and suggestions.

Yours sincerely,

The authors

Reviewer 3 Report

The article is interesting and presents model research with a border effect on yield and yield components of five varieties of wheat.

I propose to use ha instead hm2. The available nitrogen content was 86.3 g · kg-1?  perhaps 86,3 mg!! in Figure 1 total precipitation should be shown in bars and temperature - inline form (because the temperature is a function of time). How many P and N per ha were used before tillage? Why so a big amount of urea (375 kg = 172,5 kg of N) used before sowing and were the N fertilizers or N divided in 2 or 3 rates used in springtime? In Table 1 is no. of panicles - - it should be no. of ears? There are only 11 literature items - could be more!  

For me will be interesting how many ears are from one wheat plant?

Author Response

Response to comments

Dear Reviewer

We are grateful to the Journal of Agronomy for providing us with the valuable opportunity to revise and modify our manuscript. We would like to thank the reviewers for their careful review and comments, and for your hard work. All the comments have been carefully revised and highlighted in red color for your convenience. We shall look forward to hearing from you and hope that you will consider our work again and give us the opportunity to communicate further with the reviewers as needed.

Response to reviewer comments:

  1. I propose to use ha instead hm2.

Response: Thanks for pointing out the flaw in the manuscript, We have changed all the hm2 in this article to ha.

  1. The available nitrogen content was 86.3 g · kg-1?  perhaps 86,3 mg!!

Response: Thanks for pointing out the flaw in the manuscript, We have modified this part.“The soil organic matter content in the 0-20 cm soil layer of the experimental field was 18.03 g · kg-1, the total nitrogen content was 1.31 g · kg-1, the available nitrogen content was 86.3 mg · kg-1, and the available potassium was 227.48 mg · kg-1.” (Lines 78-79) 

  1. in Figure 1 total precipitation should be shown in bars and temperature - inline form (because the temperature is a function of time).

Response: Thanks for pointing out the flaw in the manuscript, We have modified Figure1. (Lines 80-81) 

  1. How many P and N per ha were used before tillage? Why so a big amount of urea (375 kg = 172,5 kg of N) used before sowing and were the N fertilizers or N divided in 2 or 3 rates used in springtime?

Response: Thanks for pointing out the flaw in the manuscript, We have modified this part.“The compound fertilizer ( N-P2O5-K2O : 24-15-5 ) was uniformly applied in the form of base fertilizer at 375 kg· ha-1 before tillage.” (Lines 99-100) 

  1. In Table 1 is no. of panicles - - it should be no. of ears?

For me will be interesting how many ears are from one wheat plant?

Response: Thanks for pointing out the flaw in the manuscript,We modified and supplemented Table 1, and then increased and calculated the effective spike number of wheat per hole.“The number of effective panicles per hole of XN175 was the largest, 23, and the number of effective panicles per plant was 2. ” (Lines 190-196) 

  1. There are only 11 literature items - could be more!

Response: Thanks for pointing out the flaw in the manuscript,We increased the number of references to 30.

In conclusion, we have tried our best to improve the manuscript and have made comprehensive changes in this paper. After reading the latest version of our manuscript, we hope you will have a better understanding of this manuscript and look forward to your evaluation. We earnestly appreciate the reviewers’ work and we hope that the changes and corrections will meet with your approval. Once again, thank you very much for your comments and suggestions.

Yours sincerely,

The authors

Round 2

Reviewer 2 Report

Thank you the authors for all answers to my suggestions. In fact the manuscript has been substantially improved. However this manuscript should not be publish in the journal Agronomy (MDPI) in carrent form. The authors should make substantial improvement. As I stated in my first opinion, the main problem of this manuscript is the lack of comparison of the effects of the new method of sowing with traditional sowing. So, if there is not such comparison, the authors couldn’t write: "....wheat suitable for hole sowing as a sowing method can maximize its performance and obtain higher yield" . Detailed suggestions are pointed in the manuscript attached.

Author Response

Response to comments

Dear Reviewer

  We are very grateful for your careful review and comments, thank you for your hard work. All reviews have been carefully modified and highlighted in red for your convenience.

Response to reviewer comments:

  Response: Thanks for pointing out the flaw in the manuscript, Our previous work and research have compared the hole sowing of wheat with drill sowing and wide sowing methods, and concluded that wheat can exert more advantages than the other two sowing methods under hole sowing conditions. Therefore, the main purpose of this experiment is to explore whether different wheat varieties are different under the condition of hole sowing, and to explore which varieties can have significant border effect under hole sowing, so as to provide further basis and reference for future work. Among the five varieties, ' XN175 ' and ' XN765 ' have the most significant border effect. Because this experiment is only through the plot test, the yield of wheat varieties with significant border effect is not necessarily the largest in the plot test. However, if it is put into actual production, due to the strong border effect, some wheat varieties can maximize the border advantage under the condition of hole sowing, thereby increasing the yield and increasing the benefit.

  We sincerely thank you for your work and we hope that these changes and corrections will be recognized by you. Thank you again for your comments and suggestions.

Yours sincerely,

The authors